# Association between Serum Levels of Interleukin-25/Thymic Stromal Lymphopoietin and the Risk of Exacerbation of Chronic Obstructive Pulmonary Disease

**DOI:** 10.3390/biom13030564

**Published:** 2023-03-20

**Authors:** Joon Young Choi, Tae-Hyung Kim, Sung-Yoon Kang, Hye Jung Park, Seong Yong Lim, Sang Hyuk Kim, Ki-Suck Jung, Kwang Ha Yoo, Hyoung Kyu Yoon, Chin Kook Rhee

**Affiliations:** 1Division of Pulmonary and Critical Care Medicine, Department of Internal Medicine, Incheon St. Mary’s Hospital, The Catholic University of Korea, Seoul 06591, Republic of Korea; tawoe@naver.com; 2Division of Pulmonary and Critical Care Medicine, Department of Internal Medicine, Hanyang University Guri Hospital, Hanyang University College of Medicine, Guri 11923, Republic of Korea; 3Division of Pulmonology and Allergy, Department of Internal Medicine, Gachon University Gil Medical Center, Incheon 22318, Republic of Korea; 4Department of Internal Medicine, Gangnam Severance Hospital, Yonsei University College of Medicine, Seoul 06273, Republic of Korea; 5Division of Pulmonary and Critical Care Medicine, Department of Medicine, Kangbuk Samsung Hospital, Sungkyunkwan University School of Medicine, Seoul 03181, Republic of Korea; 6Division of Pulmonary and Critical Care Medicine, Department of Internal Medicine, Dongguk University Gyeongju Hospital, Dongguk University College of Medicine, Gyeongju 38067, Republic of Korea; 7Department of Internal Medicine, College of Medicine, Hallym University Sacred Heart Hospital, Anyang 14068, Republic of Korea; 8Division of Pulmonary and Critical Care Medicine, Department of Internal Medicine, Konkuk University Medical Center, Konkuk University School of Medicine, Seoul 05029, Republic of Korea; 9Division of Pulmonary and Critical Care Medicine, Department of Internal Medicine, Yeouido St. Mary’s Hospital, The Catholic University of Korea, Seoul 06591, Republic of Korea; 10Division of Pulmonary and Critical Care Medicine, Department of Internal Medicine, Seoul St. Mary’s Hospital, The Catholic University of Korea, Seoul 06591, Republic of Korea

**Keywords:** alarmin, IL-25, TSLP, chronic obstructive pulmonary disease

## Abstract

Th2 inflammation is associated with various characteristics of patients with chronic obstructive pulmonary disease (COPD). In this study, we analyzed the COPD exacerbation risk associated with serum levels of interleukin (IL)-25/thymic stromal lymphopoietin (TSLP) and eosinophils. We studied the KOCOSS cohort, a multicenter COPD cohort created by 54 medical centers in South Korea. We extracted data collected between April 2012 and August 2020. We measured serum levels of TSLP and IL-25 in those who agreed to provide blood, and assessed exacerbation risk according to each. In all, 562 patients were enrolled. The IL-25-high group had a lower St. George’s Respiratory Questionnaire score than others, and the TSLP-high group had a poorer exercise capacity than the TSLP-low group. There were no significant differences in the forced expiratory volume in 1 s (FEV1), the levels of Th2 inflammatory biomarkers, or the exacerbation histories between the two groups. The 3-year decline in FEV1 was not significantly affected by IL-25 or TSLP levels. In terms of 1-year exacerbation risk, individuals in the IL-25-high group were at lower risk for moderate-to-severe exacerbation than others. A high TSLP level was associated with a lower risk of severe exacerbation but only in the eosinophil-low group. Serum levels of IL-25 are negatively correlated with moderate-to-severe exacerbation risk in this cohort. A negative correlation between severe exacerbation risk and TSLP level was apparent only in the eosinophil-low group.

## 1. Introduction

Chronic obstructive pulmonary disease (COPD) is a heterogeneous condition characterized by persistent respiratory symptoms and fixed airway obstruction attributable to airway and/or alveolar damage caused by noxious particles or gases [1]. Most COPD patients evidence immunity of types 1 and 3 involving macrophages, neutrophils, CD8+ cytotoxic T cells, CD4+ TH1 cells, TH17 cells, ILC3 cells, and B cells. These reflect the mucosal inflammatory response to inhaled irritants [2]. However, 20–40% of COPD patients present with an eosinophilic phenotype associated with type 2 inflammation [3,4]. Such patients exhibit frequent exacerbations and higher mortality rates than non-eosinophilic patients [5,6,7]. Monoclonal antibodies targeting interleukin (IL)-5 or IL-5Rα in eosinophilic COPD patients show promise in terms of reducing exacerbation [8,9,10].

Alarmins, including thymic stromal lymphopoietin (TSLP), IL-33, and IL-25, are cytokines secreted by bronchial epithelial cells in response to endogenous or exogenous danger signals. These cytokines are involved upstream of the Th2 inflammatory pathway [2]. Recently, high IL-33 levels were reported to be associated with chronic bronchitis and frequent exacerbation of COPD patients [11,12]; blockage of the IL-33 and ST2 pathways showed promise in terms of reducing COPD exacerbation [13,14]. The clinical outcomes associated with the levels of other alarmins (TSLP and IL-25) have been less thoroughly investigated. Particularly, because the anti-TSLP antibody tezepelumab showed promise in patients with severe uncontrolled asthma, biologic agents targeting the alarmins have become of increasing interest [15].

In this study, we explored the clinical characteristics of COPD according to IL-25 and TSLP levels. The exacerbation frequency over 1 year was prospectively determined by cytokine levels. As the cytokines act upstream of the Th2 inflammatory pathway, we stratified patients by high and low blood levels of eosinophils. We assessed differences in the rate of decline of the 3-year annual FEV1 according to each marker.

## 2. Materials and Methods

### 2.1. Study Population

The Korea COPD Subgroup Study (KOCOSS) cohort is a nationwide prospective group formed by 54 medical centers in South Korea, commencing in April 2012. Those aged >40 years with irreversible airflow obstructions (post-bronchodilator (BD) forced expiratory volume in 1 s (FEV1)/forced vital capacity (FVC) ≤70% of the normal predicted value) were included. Patients were evaluated at 6-month intervals after initial baseline examination. We extracted KOCOSS data from April 2012 to August 2020. We selected patients who gave blood samples. We excluded those with <10 pack-years of smoking history.

### 2.2. Clinical Characteristics

We collected baseline characteristics including age, sex, smoking history, and body mass index (BMI). Symptoms, health-related quality of life, and exercise capacity were evaluated using the modified Medical Research Council (mMRC) score, St. George’s Respiratory Questionnaire (SGRQ), the COPD assessment test (CAT), and the 6 min walk test (6MWT). To explore asthma overlap and Th2 inflammation status, we evaluated asthma history and physician-diagnosed asthma–COPD overlap (ACO). We retrieved eosinophil counts, immunoglobulin (Ig)-E levels, and fractional exhaled nitric oxide (FeNO) values.

Pulmonary function tests (PFTs), including flow–volume curve construction, lung volume measurement, and assessment of the diffusion capacity of the lungs for carbon monoxide (DLCO) were performed at enrollment and annually thereafter. We extracted 3-year PFT results to derive lung function trajectories according to IL-25/TSLP levels. We recorded exacerbations in the year before baseline evaluation and during a 1-year follow-up. Exacerbation was defined as acute deterioration of respiratory symptoms requiring additional treatment such as antibiotics or corticosteroids. Exacerbations thus treated in outpatient clinics were defined as moderate; those that required hospitalization or an emergency room visit were defined as severe.

### 2.3. Measurement of IL-25 and TSLP

Plasma levels of IL-25 and TSLP were measured using Luminex-Multiplex kits (Komabiotech, Seoul, Korea). Blood samples were centrifuged for 10 min at 1000× *g* within 30 min of collection in tubes with the anticoagulant sodium ethylenediaminetetraacetic acid (EDTA). Plasma was stored at ≤–80 °C and prepared for analysis in a 96-well plate employing a custom Human Cytokine/Chemokine Magnetic Bead Panel (Millipore Corp., Billerica, MA, USA). We followed the kit-specific protocol of the manufacturer of the Luminex 200 analyzer (Luminex, Austin, TX, USA) that ran MasterPlex QT software version 4 (MiraiBio, San Bruno, CA, USA). Standard curves for all analytes were generated using standards provided by the manufacturer. We distinguished high/low-IL-25 and high/low-TSLP groups according to median values. The median level of IL-25 was 1.33 ng/mL. We divided patients into IL-25-low and IL-25-high groups using this as the cutoff. The median TSLP level was 0.78 pg/mL; we formed TSLP-low and TSLP-high groups using this value as the cutoff.

### 2.4. Statistical Analysis

All statistical analyses were performed using R software (ver. 3.6.3; the R Foundation for Computing, Vienna, Austria). Continuous variables are shown as means ± standard deviations and categorical variables are given as numbers (percentages). We compared the clinical parameters of each group, and performed subgroup analyses according to eosinophil levels (≥300 vs. <300 cells/mm^3^). Groups were compared using the Pearson χ^2^ test for categorical variables and the Student’s t-test for continuous variables. Correlations between blood eosinophil counts and cytokine levels were evaluated by deriving Pearson correlation coefficients. The 3-year rates of decline in FEV1 were compared using a linear mixed model adjusted for age, sex, and BMI as covariates. The exacerbation risk for each group was analyzed using a binomial regression model after adjustment for age and smoking and exacerbation histories. A *p*-value < 0.05 was considered statistically significant.

## 3. Results

### 3.1. Baseline Characteristics

We included 562 patients; Table 1 summarizes their baseline characteristics. The mean age was 68.8 ± 7.6 years and most were male (96.8%). Of all patients, 29.5% were current smokers with mean BMI 23.2 ± 3.2 kg/m^2^. The mean mMRC, SGRQ, and CAT scores were 1.1 ± 0.9, 30.4 ± 18.7, and 14.0 ± 7.9, respectively, and the mean 6MWT distance was 409.6 ± 106.7 m. Of all patients, 23.6% had histories of asthma and 21.0% had been physician-diagnosed with ACO. The mean blood eosinophil count, IgE level, and FeNO values were 226.8 ± 250.2 cells/mm^3^, 216.5 ± 247.0 mg/dL, and 26.8 ± 17.1 ppb, respectively. The mean post-BD FEV1% predicted and FEV1/FVC were 64.9 ± 18.7% and 52.4 ± 11.5, respectively. Of all patients, 12.9% had a history of exacerbation in the year prior to enrollment.

### 3.2. Clinical Characteristics of the IL-25-Low and IL-25-High Groups

The clinical parameters according to different levels of IL-25 are shown in Table 2. There were no significant differences in age, sex, smoking history, or BMI between the two groups. There were also no significant differences in symptoms or exercise capacity, except that the SGRQ score was higher in the low group. There were no between-group differences in lung function, the levels of Th2 inflammatory biomarkers, or the histories of prior exacerbations.

### 3.3. Clinical Characteristics of the TSLP-Low and TSLP-High Groups

The clinical parameters according to different levels of TSLP are shown in Table 3. The TSLP-low group was younger (67.8 ± 7.4 vs. 69.4 ± 7.7 years, *p* = 0.02) and individuals in the group were more likely to be current smokers (34.9% vs. 6.3%, *p* = 0.04). There were no differences in the symptom scores; however, the 6MWT exercise capacity was poorer in the TSLP-high group (422.7 ± 106.0 vs. 400.6 ± 106.5 m, *p* = 0.03). The FEV1 did not differ between the two groups; however, the FEV1/FVC and DLCO were lower in the TSLP-high group (53.6 ± 11.4 vs. 51.6 ± 11.6, *p* = 0.04; 67.7 ± 18.0 vs. 63.1 ± 18.4%, *p* = 0.01, respectively). There were no differences in the levels of Th2 inflammatory biomarkers or the numbers of prior exacerbations. Also, the differences in the clinical characteristics according to the breakdown of IL-25 and TSLP values are shown in Appendix A. 

### 3.4. Differences in Decline Trajectories of FEV1

There were no significant differences in the rate of decline in FEV1 according to IL-25 (Figure 1) or TSLP level (Figure 2).

### 3.5. Differences in Exacerbation Risk

Overall, high levels of IL-25 were associated with a lower risk of future moderate-to-severe exacerbation (OR = 0.69, 95% CI [0.49–0.98]); high TSLP levels had no such association (Table 4). In subgroup analyses according to eosinophil level, neither the IL-25 level nor TSLP level significantly affected future exacerbation risk in the eosinophil-high group. However, in the eosinophil-low group, a high TSLP level was significantly associated with a lower risk of future severe exacerbation (OR = 0.32, 95% CI [0.11–0.92]). Moreover, risk of moderate-to-severe exacerbation were comparable between patients with IL-25 high & TSLP high group, compared to those with both low group (Appendix A). 

## 4. Discussion

We explored the clinical characteristics of COPD patients according to IL-25 and TSLP levels. Patients with higher IL-25 levels evidenced a better health-related quality of life; patients with higher TSLP levels exhibited poorer exercise capacity. There were no significant differences in lung function, the rate of decline in FEV1, or exacerbation history between the low- and high-cytokine groups. Although IL-25 and TSLP initiate Th2 inflammation in response to various antigens or allergens, the biomarkers of such inflammation (blood eosinophil count, as well as IgE and FeNO levels) were comparable among groups. However, and importantly, the risk of moderate-to-severe exacerbation was lower in the IL-25-high group and the risk of severe exacerbation was lower in TSLP-high patients with lower blood eosinophil counts.

IL-25, TSLP, and IL-33 are alarmins produced by bronchial epithelial cells in response to exogenous or endogenous danger signals [16]. These cytokines act upstream of the Th2 inflammatory cascade, thus initiating the Th2 response and eosinophilic inflammation caused by Th2 lymphocytes and type 2 innate lymphoid cells that produce cytokines including IL-4, IL-5, and IL-13 [2,17,18]. COPD is a heterogeneous disease; the principal pathophysiologies are immunity of types 1 and 3 [2]. However, in some COPD patients, Th2 airway inflammation is important; such patients include those with eosinophilic COPD or ACO [2,19,20].

About 20–40% of COPD patients show eosinophilic inflammation [3,4]. Clinical outcomes according to eosinophil levels have varied among studies. In one previous study, 49% of COPD patients were persistently eosinophilic and 47% were intermittently eosinophilic using a cutoff of 2% [21]. Those with persistent eosinophilia had a higher FEV1 and better health-related quality of life than the others. There were no significant differences in longitudinal changes in the FEV1 or COPD exacerbations. By contrast, another study reported that eosinophil levels > 340 cells/mm^3^ were associated with a 1.76-fold increased risk of severe exacerbation [7]; another study reported a 1.49-fold increased risk of exacerbation in eosinophilic COPD patients in the KOCOSS database and accelerated rate of decline in lung function in non-eosinophilic COPD patients [6]. However, one single-center retrospective study reported opposite results [22]. Finally, a large population-based study in the Netherlands reported that blood eosinophilia (≥275 cells/mm^3^) was associated with increased all-cause mortality [5].

TSLP is an innate cytokine that plays important roles in allergic and adaptive airway inflammation [23]. Its expression is elevated in asthmatic patients and is inversely correlated with lung function [24,25]. It is also elevated in COPD patients, and negatively correlated with eosinophil levels in sputum. Thus, TSLP may also be associated with non-allergic immunity [16,26]. Higher expression of TSLP in certain COPD patients may be associated with cigarette smoke, which induces its expression in the epithelium [27,28]. The potential role of IL-25 in COPD has received less attention. IL-25 levels increase in COPD patients with high TSLP levels [29]. Its level can be used to distinguish COPD from asthma, being significantly higher in the latter [30]. Surprisingly, we found that the levels of both IL-25 and TSLP were negatively correlated with the risk of future exacerbation, but the blood levels of eosinophils and IL-33 were positively correlated in previous studies [5,11]. Further work on these different pathophysiologies is required, as is an exploration of whether biological therapies would usefully target certain cytokines.

IL-33, another alarmin, is a member of the IL-1 family that signals via the ST2 pathway to promote Th2 inflammation [12]. IL-33 and ST2 levels are increased in COPD patients, and their expression is induced by cigarette smoke [31]. A high IL-33 level is associated with a chronic bronchitis phenotype and increased risk of acute exacerbation [11,12]. Recently, blockage of this pathway has yielded promising results. Itepekimab, a double-blind phase 2a trial, treated former smokers with COPD with monoclonal antibodies targeting IL-33; treatment reduced the risk of exacerbation and increased lung function [13]. In another study, astegolimab, a selective ST2 IgG2 monoclonal antibody, improved the health-related quality of life of patients with moderate-to-very-severe COPD [14]. Recently, tezepelumab (an anti-TSLP antibody) was found to lower the exacerbation risk, increase the FEV1, and enhance asthma control and health-related quality of life in a phase 3 trial in patients with severe uncontrolled asthma [15]. An ongoing clinical trial (NCT04039113) is investigating a possibly useful role in COPD patients. However, the decreased exacerbation risk that we found in the TSLP-high eosinophil-low group may not reflect blockage of the IL-33 and ST2 pathways.

Our work has certain limitations. First, most patients were male. However, no previous report has described any sex differences in IL-25 or TSLP levels. Second, most patients were treated in tertiary hospitals; they do not represent the entire COPD population. Third, we only collected the drug history of patients at baseline, and therefore it is unknown whether treatment affected the levels of IL-25 and TSLP. The most novel finding of our study was the inverse correlations between IL-25/TSLP and exacerbation risk, which makes it surprising that another alarmin, IL-33, was shown to have a positive correlation. Our work is the first to analyze the differences of IL-25/TSLP and clinical characteristics (including exacerbation risks) in COPD patients.

## 5. Conclusions

We investigated the roles of alarmins in COPD patients. A high IL-25 level was associated with a favorable health-related quality of life, and a high TSLP level was associated with poor exercise capacity. Unlike IL-33 and eosinophil count, IL-25 and TSLP levels were inversely associated with exacerbation risk, although the association between TSLP level and reduced exacerbation risk was significant only in those with low eosinophil counts. Thus, IL-25 and TSLP are potential biomarkers of prognosis and future targets for use in biologic therapy.

## Figures and Tables

**Figure 1 biomolecules-13-00564-f001:**
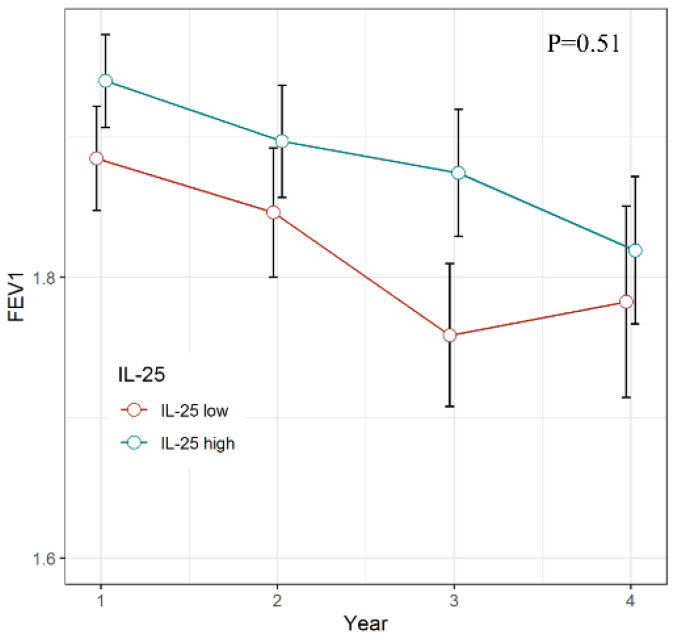
Differences in lung function trajectory between IL-25-low and IL-25-high groups.

**Figure 2 biomolecules-13-00564-f002:**
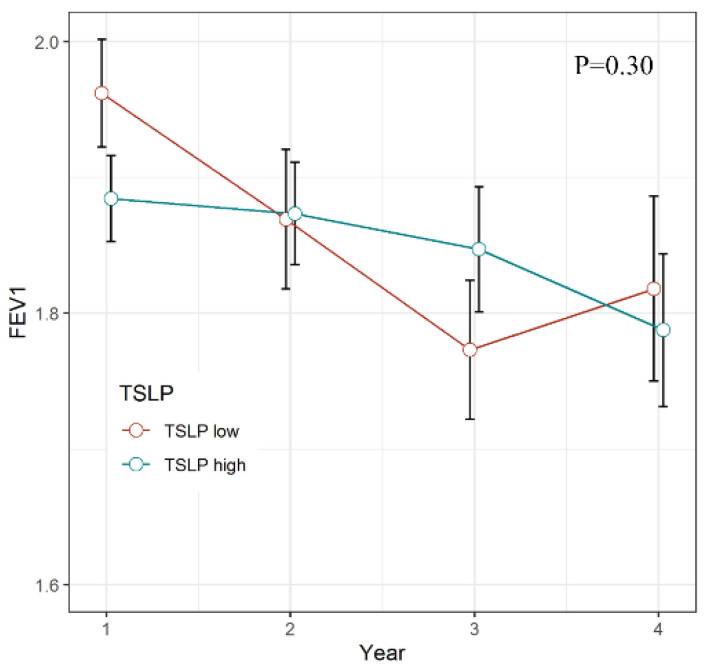
Differences in lung function trajectory between TSLP-low and TSLP-high groups.

**Table 1 biomolecules-13-00564-t001:** Baseline characteristics.

	Overall (*n* = 562)
Age	68.8 ± 7.6
Sex (male)	544 (96.8%)
Smoking history	
- Former smoker	396 (70.5%)
- Current smoker	166 (29.5%)
BMI (kg/m^2^)	23.2 ± 3.2
mMRC score	1.1 ± 0.9
SGRQ score	30.4 ± 18.7
CAT score	14.0 ± 7.9
6MWT (m)	409.6 ± 106.7
Hx of asthma	131 (23.6%)
Physician-diagnosed ACO	97 (21.0%)
GOLD stage	
I	87 (15.5%)
II	333 (59.4%)
III	108 (19.3%)
IV	33 (5.9%)
FEV1 (L)	1.9 ± 0.6
FEV1 (%)	64.9 ± 18.7
FVC (L)	3.6 ± 0.7
FVC (%)	85.5 ± 15.1
FEV1/FVC	52.4 ± 11.5
DLCO (mL/mmHg/min)	64.9 ± 18.4
Blood eosinophil count (/mm^3^_)_	226.8 ± 250.2
IgE (mg/dL)	216.5 ± 247.0
FeNO (ppb)	26.8 ± 17.1
Hx of exacerbation	71 (12.9%)
IL-25 (ng/mL)	1.3 ± 0.4
TSLP (pg/mL)	0.8 ± 0.7

Data are presented as *n* (%) or mean ± SD. BMI, body mass index; mMRC, modified Medical Research Council; SGRQ, St. George’s Respiratory Questionnaire; CAT, COPD assessment test; 6MWT, 6 min walk test; Hx, history; ACO, asthma–COPD overlap; FEV1, forced expiratory volume in 1 s; FVC, forced vital capacity; DLCO, diffusion capacity of lungs for carbon monoxide; IgE, immunoglobulin E; FeNO, fractional exhaled nitric oxide.

**Table 2 biomolecules-13-00564-t002:** Differences in the clinical characteristics between IL-25-low/high groups.

	IL-25-Low (*n* = 212)	IL-25-High (*n* = 350)	*p*-Value
Age	68.6 ± 7.8	68.9 ± 7.5	0.64
Sex (male)	256 (97.3%)	288 (96.3%)	0.66
Smoking history			0.48
- Former smoker	181 (68.8%)	215 (71.9%)	
- Current smoker	82 (31.2%)	84 (28.1%)	
BMI (kg/m^2^)	23.2 ± 3.3	23.1 ± 3.1	0.77
mMRC score	1.1 ± 0.8	1.2 ± 0.9	0.37
SGRQ score	33.0 ± 19.6	28.1 ± 17.5	<0.01
CAT score	14.4 ± 8.0	13.5 ± 7.8	0.18
6MWT (m)	401.2 ± 102.6	417.2 ± 110.0	0.11
Hx of asthma	56 (21.6%)	75 (25.4%)	0.34
Physician-diagnosed ACO	48 (22.2%)	49 (19.9%)	0.62
GOLD stage			0.136
I	40 (15.2%)	47 (15.8%)	
II	151 (57.4%)	182 (61.1%)	
III	50 (19.0%)	58 (19.5%)	
IV	22 (8.4%)	11 (3.7%)	
FEV1 (L)	1.9 ± 0.6	1.9 ± 0.6	0.33
FEV1 (%)	64.2 ± 19.6	65.6 ± 17.9	0.39
FVC (L)	3.5 ± 0.8	3.6 ± 0.7	0.49
FVC (%)	85.0 ± 15.5	85.9 ± 14.8	0.48
FEV1/FVC	51.6 ± 11.8	53.1 ± 11.2	0.11
DLCO (mL/mmHg/min)	65.1 ± 18.5	64.7 ± 18.4	0.83
Blood eosinophil count (/mm^3^)	206.2 ± 175.3	245.0 ± 300.5	0.07
IgE (mg/dL)	221.3 ± 258.9	211.9 ± 235.4	0.69
FeNO (ppb)	26.1 ± 16.4	27.7 ± 18.0	0.62
Hx of exacerbation	28 (10.9%)	43 (14.7%)	0.23
IL-25 (ng/mL)	1.2 ± 0.1	1.5 ± 0.5	<0.01
TSLP (pg/mL)	0.8 ± 0.4	0.9 ± 0.8	0.08

Data are presented as *n* (%) or mean ± SD. BMI, body mass index; mMRC, modified Medical Research Council; SGRQ, St. George’s Respiratory Questionnaire; CAT, COPD assessment test; 6MWT, 6 min walk test; Hx, history; ACO, asthma–COPD overlap; FEV1, forced expiratory volume in 1 s; FVC, forced vital capacity; DLCO, diffusion capacity of lungs for carbon monoxide; IgE, immunoglobulin E; FeNO, fractional exhaled nitric oxide.

**Table 3 biomolecules-13-00564-t003:** Differences in clinical characteristics between TSLP-low/high groups.

	TSLP-Low (*n* = 212)	TSLP-High (*n* = 350)	*p*-Value
Age	67.8 ± 7.4	69.4 ± 7.7	0.02
Sex (male)	203 (95.8%)	341 (97.4%)	0.40
Smoking history			0.04
- Former smoker	138 (65.1%)	258 (73.7%)	
- Current smoker	74 (34.9%)	92 (26.3%)	
BMI (kg/m^2^)	23.5 ± 3.2	23.0 ± 3.2	0.08
mMRC score	1.1 ± 0.9	1.2 ± 0.9	0.35
SGRQ score	31.3 ± 19.2	29.8 ± 18.3	0.33
CAT score	14.2 ± 8.0	13.8 ± 7.9	0.60
6MWT (m)	422.7 ± 106.0	400.6 ± 106.5	0.03
Hx of asthma	49 (23.8%)	82 (23.6%)	1.00
Physician-diagnosed ACO	43 (23.5%)	54 (19.4%)	0.34
GOLD stage			0.681
I	31 (14.7%)	56 (16.0%)	
II	132 (62.6%)	201 (57.4%)	
III	37 (17.5%)	71 (20.3%)	
IV	11 (5.2%)	22 (6.3%)	
FEV1 (L)	1.9 ± 0.6	1.9 ± 0.6	0.23
FEV1 (%)	65.5 ± 18.2	64.5 ± 19.1	0.54
FVC (L)	3.6 ± 0.7	3.5 ± 0.8	0.81
FVC (%)	85.5 ± 15.2	85.4 ± 15.1	0.98
FEV1/FVC	53.6 ± 11.4	51.6 ± 11.6	0.04
DLCO (mL/mmHg/min)	67.7 ± 18.0	63.1 ± 18.4	0.01
Blood eosinophil count (/mm^3^)	235.9 ± 278.4	221.4 ± 231.9	0.54
IgE (mg/dL)	213.2 ± 262.7	218.4 ± 238.	0.83
FeNO (ppb)	26.6 ± 17.2	27.0 ± 17.2	0.91
Hx of exacerbation	27 (12.9%)	44 (12.9%)	1.00
IL-25 (ng/mL)	1.3 ± 0.2	1.4 ± 0.5	0.07
TSLP (pg/mL)	0.4 ± 0.2	1.0 ± 0.8	<0.01

Data are presented as *n* (%) or mean ± SD. BMI, body mass index; mMRC, modified Medical Research Council; SGRQ, St. George’s Respiratory Questionnaire; CAT, COPD assessment test; 6MWT, 6 min walk test; Hx, history; ACO, asthma–COPD overlap; FEV1, forced expiratory volume in 1 s; FVC, forced vital capacity; DLCO, diffusion capacity of lungs for carbon monoxide; IgE, immunoglobulin E; FeNO, fractional exhaled nitric oxide.

**Table 4 biomolecules-13-00564-t004:** Exacerbation risk according to IL-25 and TSLP.

	Moderate-to-Severe Exacerbation			Severe Exacerbation		
	OR	95% CI	*p*-Value	OR	95% CI	*p*-Value
Overall						
IL-25-high	0.69	0.49–0.98	0.04	0.08	0.20–1.12	0.08
TSLP-high	1.02	0.70–1.47	0.93	0.48	0.21–1.08	0.08
Eosinophil-high						
IL-25-high	0.64	0.34–1.19	0.16	0.65	0.20–2.10	0.46
TSLP-high	1.14	0.58–2.26	0.69	0.95	0.29–3.39	0.93
Eosinophil-low						
IL-25-high	0.66	0.43–1.00	0.05	0.41	0.00–1.18	0.11
TSLP-high	0.90	0.58–1.40	0.64	0.32	0.11–0.92	0.04

Adjusted covariates: age, smoking history, past exacerbation history. IL-25, interleukin-25; TSLP, thymic stromal lymphopoietin.

## Data Availability

The datasets supporting the conclusions of this article cannot be shared publicly because the data of the KOCOSS study are third party data. Data are available from the KOCOSS (www.kocoss.kr, accessed on 1 July 2022) for researchers who meet the criteria for access to confidential data.

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
