# Peer review of "Association between Serum Levels of Interleukin-25/Thymic Stromal Lymphopoietin and the Risk of Exacerbation of Chronic Obstructive Pulmonary Disease"

_biomolecules, 2023, doi:10.3390/biom13030564_

Round 1

Reviewer 1 Report

The authors have assessed clinical characteristics of COPd based on IL25 and TSLP levels. They have used standardized questionnaires to collect information and performed analyses. The authors evaluated the plasma levels of IL25 and TSLP and correlated their findings from the questionnaire to hi and low levels of these alarmins. The study i presented in a well organized manner.

Major suggestions:

1. The results of the IL25 and TSLP multiplex assay are not provided.

2. the results only focus on high and low values of IL25 and TSLP. were there any overlaps in IL25 and TSLP high and low patients? ie were there any IL25-high/TSLP-high and IL25-low/TSLP- low ? if so how did that affect the outcome of the study?

3. The discussion should also highlight the novelty of this study which is currently lacking.

Author Response

1. The results of the IL25 and TSLP multiplex assay are not provided.

A) Thank you for your valuable comment. We have conducted additional analysis by calculating the level of IL-25 and TSLP in the overall population and as well as differences between high and low levels of IL-25/TSLP. The results of this analysis are presented in Table 1-3.

2. The results only focus on high and low values of IL25 and TSLP. were there any overlaps in IL25 and TSLP high and low patients? ie were there any IL25-high/TSLP-high and IL25-low/TSLP- low ? if so how did that affect the outcome of the study?

A) We have further analyzed the differences of clinical characteristics according to the breakdown of IL-25 and TSLP values. There were significant differences in the SGRQ score and 6MWT between groups. SGRQ score was poorest in the both low group and most favorable in both high group. We added these information in the supplementary material (Table S1). The differences of moderate-to-severe exacerbation frequency was not significant between both high and both low group (Table S2).

3. The discussion should also highlight the novelty of this study which is currently lacking.

A) Most novel finding of our study was the inverse correlations between IL-25/TSLP and exacerbation risk; which is surprising that another alarmin, IL-33, was shown to be positive correlation. Our work was the first to analyze the differences of IL-25/TSLP and clinical characteristics (including exacerbation risks) in the COPD patients.

We have added these sentences in the discussion (page 13, line 276-279)

Reviewer 2 Report

The design of the article is well done. The number of volunteers is quite good. Exacerbation criteria can be specified more clearly. It would be more appropriate not to use the exacerbation data of the previous year.  Tables and graphics are sufficient and understandable.  Grammar and spelling errors should be corrected.

Author Response

The design of the article is well done. The number of volunteers is quite good. Exacerbation criteria can be specified more clearly. It would be more appropriate not to use the exacerbation data of the previous year.  Tables and graphics are sufficient and understandable.  Grammar and spelling errors should be corrected.

A) Thank you for your valuable comments. Exacerbation data of the previous year was only used as adjustment variable in multivariate analysis. We did not use exacerbation data of previous year as outcome of this study. For the exacerbation analysis, we have utilized prospectively collected exacerbation data.

 The English in this document has been checked by at least two professional editors, both native speakers of English. For a certificate, please see:  http://www.textcheck.com/certificate/wK3aal

If there are other grammar and spelling errors to be corrected, please let us know.

Reviewer 3 Report

The manuscript is well-written and interesting to read. A few comments:

1. The participant group classification details are not included in the Methods, although the authors mentioned the group in the results. It will be easier to know the participant group design in the method. And what is the COPD severity classification in this study?

2. The clinical characteristics analysis of IL-25 and TSLP groups needs more analysis and explanation. The authors compared all characteristics between low/high groups. However, these comparisons didn't explain whether the low/high level correlated to COPD severity. Does the COPD severity affect the IL-25 and TSLP levels?

3. Does the treatment affect the level of IL-25 and TSLP?

4. In the discussion, the authors mentioned TSLP levels could be used to distinguish COPD from asthma. How about ACO patients?  We know that ACO is another clinical group of patients, but still no clear diagnostic standards for it. In the study, the authors included the ACO participant data; how does the study findings can implement into clinical use?

Author Response

1. The participant group classification details are not included in the Methods, although the authors mentioned the group in the results. It will be easier to know the participant group design in the method. And what is the COPD severity classification in this study?

A) We moved the cut-off value of IL-25 high/low group and TSLP high/low group from the result section to method section (page 3, line 112-115). We have classified the group according to the median value. It is because there has been no previous reference regarding the cut off level of IL-25 and TSLP in blood sample of COPD.

The COPD severity classification in our study was as below:

  • Mild (GOLD I, FEV1≥80%): 87(15.5%)
  • Moderate (GOLD II, FEV1 50-80%): 333 (59.4%)
  • Severe (GOLD III, FEV1 30-50%): 1087 (19.3%)
  • Very severe (GOLD IV, FEV1<30%): 33 (5.9%)

We added this information at the Table 1.

2. The clinical characteristics analysis of IL-25 and TSLP groups needs more analysis and explanation. The authors compared all characteristics between low/high groups. However, these comparisons didn't explain whether the low/high level correlated to COPD severity. Does the COPD severity affect the IL-25 and TSLP levels?

A) We have further analyzed differences of COPD severity according to IL-25 and TSLP level.

There were no significant differences of COPD severity between groups.

We added this information at the Table 2-3

3. Does the treatment affect the level of IL-25 and TSLP?

A) Unfortunately, we only gathered information of the drugs at the baseline. To analyze treatment effect of the cytokines, it is important to know clinical parameters before and after the treatment, which our database is lacking.  We appreciate your valuable comment. We have added this limitation in discussion of revised manuscript. (page 13 line 270-272)

4. In the discussion, the authors mentioned TSLP levels could be used to distinguish COPD from asthma. How about ACO patients?  We know that ACO is another clinical group of patients, but still no clear diagnostic standards for it. In the study, the authors included the ACO participant data; how does the study findings can implement into clinical use?

A) In the discussion, we mentioned that the level of IL-25 may be used to distinguish COPD from asthma (page 12, line 246-247). However, this was based on the previous study by Katoh et al.(https://doi.org/10.1080/02770903.2017.1391281) which was a small single-center study (n=60 for asthma, 30 for COPD) that needs to be more validated.

We have performed additional analysis according to the reviewer's comment. Unfortunately, there was no significant difference in the level of IL-25 or TSLP between COPD and ACO. Currently, we do not have sample in asthma only patients. To answer the question of the reviewer, we need to collect sample from pure asthma patients. Thank you very much for your comment.